# ClozEx: A Task toward Generation of English Cloze Explanation

**Zizheng Zhang** ♠ and **Masato Mita** ♠♣ and **Mamoru Komachi** ♠♡

♠ Tokyo Metropolitan University. 6-6 Asahigaoka, Hino, Tokyo 191-0065, Japan
♣ CyberAgent, Inc. 2-24-12 Shibuya Shibuya-ku, Tokyo 150-6121, Japan
♡ Hitotsubashi University. 2-1 Naka, Kunitachi, Tokyo 186-8601, Japan
zhang-zizheng@ed.tmu.ac.jp, mita_masato@cyberagent.co.jp,
mamoru.komachi@r.hit-u.ac.jp

## Abstract

Providing explanations for cloze questions in language assessment (LA) has been recognized as a valuable approach to enhancing the language proficiency of learners. However, there is a noticeable absence of dedicated tasks and datasets specifically designed for generating language learner explanations. In response to this gap, this paper introduces a novel task ClozEx of generating explanations for cloze questions in LA, with a particular focus on English as a Second Language (ESL) learners. To support this task, we present a meticulously curated dataset comprising cloze questions paired with corresponding explanations. This dataset aims to assess language proficiency and facilitates language learning by offering informative and accurate explanations. To tackle the task, we fine-tuned various baseline models with our training data, including encoder-decoder and decoder-only architectures. We also explored whether large language models (LLMs) are able to generate good explanations without fine-tuning, just using pre-defined prompts. The evaluation results demonstrate that encoder-decoder models have the potential to deliver fluent and valid explanations when trained on our dataset. [1]

## 1 Introduction

Cloze questions (Taylor, 1953) are a fundamental component of language assessment (LA). They typically consist of a sentence or a passage with certain words or phrases omitted, and language learners are required to select or fill in the most appropriate word to complete the text. Cloze questions in language educational settings are usually used in language educational settings to evaluate language proficiency in terms of various aspects such as grammatical knowledge (Rye, 1982; Alderson, 1979) and reading comprehension skills (Raymond,

---

[1] Dataset and codes are available at https://github.com/zz-zhang/ClozEx.

| Question: |
|---|
| The work was done ____ the Rehabilitation Institute of Chicago under an $8-million grant from the Army. |
| (A) at (B) down (C) round (D) of |
| **Explanation**: |
| Based on the context of the sentence, option (A) "at" is the appropriate choice for the cloze question. "At" indicates a specific location or arrival at a particular place or position, such as "he is at the store." |

Table 1: Examples of data for the ClozEx task. A system is expected to receive **Question** as input and produce **Explanation** as output.

1988; Klein-Braley, 1997). They are also widely employed in famous tests for English as a Second Language (ESL) learners, such as the International English Language Testing System (IELTS) and Test of English as a Foreign Language (TOEFL).

Explanations for cloze questions play a crucial role in language learning, particularly in self-study contexts. When learners encounter challenging cloze questions, having access to clear and concise explanations after answering the question can greatly aid their understanding of the correct answers (Williams et al., 2010). Explanations provide learners with insights into the reasoning behind the correct and incorrect choices, helping them identify and rectify their own misconceptions. The provision of high-quality explanations can empower learners, fostering deeper comprehension and long-term knowledge retention.

However, despite its usefulness, there has been almost no work on generating high-quality explanations for given cloze questions. One essential reason is that no dataset for such a task is available. Because of high costs in terms of time and human effort, employing experts to create such a dataset, to provide abundant data, is difficult, al-

though it could guarantee the quality of the dataset. Furthermore, even if a dataset could be constructed, it would be challenging to automatically generate human-like cloze explanation.

To address these challenges, we propose the task ClozEx of generating explanations for English cloze questions. Intuitively, a good explanation that helps answer a cloze question should be easy to read and provide sufficient background knowledge. Therefore, fluency and informativeness should be considered in explanation generation. We also provide a large dataset comprising over 140k expert-quality-assured (question, explanation) pairs, an example of which is shown in Table 1.

Finally, to investigate the factors that contribute to addressing the ClozEx task, we train various models as baselines, including encoder-decoder and decoder-only architectures. We also investigated the performance of large language model (LLMs) in a zero-shot prediction scenario, in which we employed LLMs to generate explanations for given cloze questions without fine-tuning. The evaluation of baseline models indicated that both encoder-decoder and decoder-only models after fine-tuning are able to produce acceptable explanations. Meanwhile, LLMs are generally good at generating fluent explanations, but in most cases, these explanations do not provide sufficient information for answering questions. Only providing LLMs with questions with naive prompts is challenging to generate high-quality explanations efficiently.

The main contributions of this work are summarized as follows:

- We propose a new task toward generation of fluent and valid English cloze explanation (ClozEx) for ESL learning.

- We create a large-scale and expert-quality-assured dataset for ClozEx task, including more than 140k instances generated by a pattern-based method.

- We investigate model performance trained on our dataset. We also explore the ability of LLMs of generating appropriate explanations in zero-shot scenario.

- We examine the correlation between automatic evaluation metrics and manual evaluation in the context of the ClozEx task, providing insights into the reliability of these metrics for assessing the quality of generated explanations.

## 2 Related Work

Owing to the efficiency of cloze questions in language assessment, previous research focused on the automatic generation of cloze questions. In contrast to early-period studies that employed naive approaches such as fixed ratio word deletion and random selection of distractor options (Rye, 1982; Bachman, 1985), current research places greater emphasis on ensuring the validity of generated questions. For example, Sakaguchi et al. (2013) created distractor options based on an English learner writing correction corpus. With this method, words that tend to be misused in a given context would be selected as the distractor options. Therefore, questions generated from their methods are expected to be more discriminative in measuring the language proficiency of learners. Additionally, previous research (Goto et al., 2010; Correia et al., 2012; Hill and Simha, 2016; Jiang et al., 2020; Panda et al., 2022) investigated what aspects of features affect the validity of cloze questions, such as part of speech (POS), $n$-gram frequency, and sense of words. They then generated questions based on such features. However, although question generation tasks do primarily aim to generate plausible questions based on given texts, they often do not explicitly address the generation of accompanying explanations. The ability to generate explanations alongside cloze questions is crucial for providing comprehensive support to language learners. By shifting the focus from generating questions to generating explanations, this research introduces a novel task that contributes to the advancement of language learning technologies.

Nagata (2019) proposed a task of feedback comment generation (FCG) for writing learning with a corresponding dataset. The FCG task automatically generates feedback comments such as a hint or an explanatory note for writing learning for non-native learners of English. However, although this task contributes to grammar learning through writing correction, it has certain limitations in facilitating systematic grammar learning. Firstly, the FCG task primarily relies on free English composition, which adopts a bottom-up approach to provide grammatical knowledge. Consequently, it inevitably lacks comprehensive coverage of grammar items that learners need to master. In contrast, cloze questions are meticulously designed by experts, adhering to established learning guidelines, thereby ensuring a certain level of coverage of grammar items that

learners should be familiar with. Secondly, the primary focus of the FCG task lies in explaining the appropriateness of specific words within a sentence, rather than elucidating why certain plausible expressions should be avoided. Furthermore, the commentaries in the FCG task stem from free composition, making it challenging to scale the production of high-quality commentaries without significant manual effort. In contrast, the `ClozEx` task builds "patterns" for each grammar item, as the cloze questions are constructed using a top-down approach. Consequently, it becomes feasible to automatically generate a considerable number of high-quality explanatory (details are explained in Section 3.2), as demonstrated by the generation of over 140k such sentences in this study.

## 3 `ClozEx` Task and Dataset

### 3.1 Task Definition

Methods devised to address the `ClozEx` task are expected to operate on a cloze question $q$ as input. A cloze question comprises a sentence with a blank, denoted as $sent$, and a set of options $OPT = [opt_1, opt_2, ..., opt_n]$ (typically, $n$ equals to 4). The objective of the methods is to generate an explanation text $exp$ as output for the given question. The generated explanation should satisfy two criteria: (1) fluency (Cotter, 2012), meaning that the explanation should be coherent and easily comprehensible, because an explanation that is difficult to read would not effectively aid language learning; (2) validity (Cross, 1991), indicating that the explanation should provide sufficient information, such as relevant language knowledge, to facilitate answering the question accurately.

### 3.2 Data Preparation

Experts in English education can be hired to write explanations for cloze questions to provide very high-quality data. However, because of the consumption of time and human effort, datasets created in such a way are scale-limited. To mitigate the considerable cost associated with manual explanation generation, we need to explore an automated method for creating both the questions and explanations in our dataset.

Experts design cloze questions in a top-down manner, starting with a specific grammatical item. Subsequently, they designed various questions based on the grammatical item (Rye, 1982). Such grammatical items could be regarded as a pattern of a specific group of cloze questions. A pattern can also be used to create new cloze questions with explanations. Thus, we designed a pattern-based method for automatic cloze question and explanation generation. This method extracts patterns from expert-designed cloze questions and explanations to ensure the quality. Then these patterns are used to generate new questions and explanations.

The data creation process is outlined in Figure 1. This method involves the extraction of patterns from expert-designed cloze questions and their corresponding explanations. These patterns serve as the foundation for generating new questions and explanations based on a publicly available corpus. During the question creation phase, sentences from a news corpus that align with a given pattern are selected. Distractor options are then generated based on which aspect of language is measured. For the explanation generation process, templates tailored to the question type are designed. These templates are populated with question and pattern information to yield initial explanations. Finally, we employ LLMs to paraphrase the template-based explanations, enhancing their fluency and diversifying their expression [2].

### 3.3 Target Types

We begin by focusing on three specific types of cloze questions: affix, verb-tense, and preposition. These question types have been selected based on their prominence in language assessment (Mochizuki and Aizawa, 2000; Collins, 2007; Chodorow et al., 2010), particularly in the context of the Test of English for International Communication (TOEIC). Affix questions require ESL learners to differentiate POS of options by analyzing prefixes or suffixes. Verb-tense questions prompt learners to identify the appropriate tense of the sentence and options. Preposition questions necessitate learners to comprehend the meaning of a sentence and consider the potential senses of the options (see Table 6 in Appendix A for examples).

The comprehension of affix and verb tense questions often relies on a narrower context within the sentence, allowing learners to answer without necessarily reading the entire sentence. By contrast, preposition questions require a comprehensive understanding of the sentence and an awareness of the various senses associated with prepositions. There-

---

[2]To avoid redundancy, or an excessive amount of irrelevant information, in the generated explanation, we set a maximum length for the explanation (128 words).

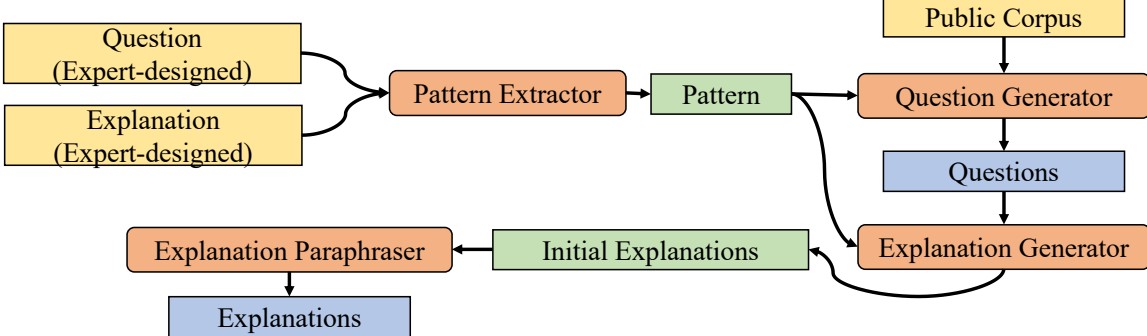

Figure 1: Pipeline of data creation method. Yellow rectangles symbolize input to the pipeline, whereas blue rectangles represent output. Modules are depicted in orange, and their corresponding intermediate results are highlighted in green.

The Westchester Philharmonic received a national award for its education program three years ago.

(a) Example of $sent^{ans}$; red word represents the answer option, and green ones denote hint words extracted from expert-designed explanation.

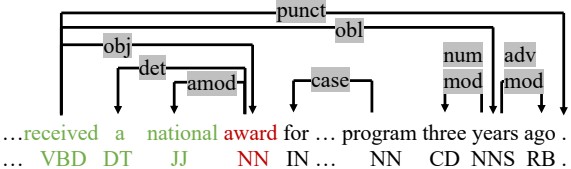

(b) Partial dependency parsing tree of $sent^{ans}$ in (a). Only nodes of colored words are extracted as $pattern$ (Pattern in Figure 1).

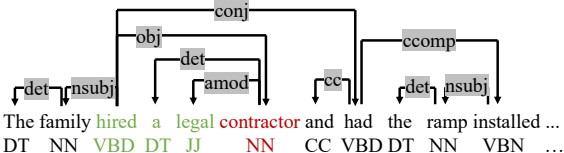

(c) Partial $\tilde{sent}_i$ and its dependency parsing tree. Because $\tilde{tree}_i$ consists of $pattern$ (marked in colored text), $\tilde{sent}_i$ could be used to generate a question.

**Question:**
The family hired a legal ______ and had the ramp installed at the front of their home at the Woodlands at Copperstone in Brentwood.
(A) contractual   (B) contractor   (C) contracted   (D) contractable
**Initial Explanation:**
The word in the blank should be the object of "hired".
"a" is the determiner of the blank. "legal" is the the adjective modifier of the blank.
Thus, a Noun, singular or mass is required.
(A) contractual is a Adjective. (B) contractor is a Noun, singular or mass. (C) contracted is a Verb, past tense. (D) contractable is a Adjective.
Therefore, the correct answer is (B) contractor.

(d) Example of generated question and corresponding initial explanation (Initial Explanations in Figure 1).

Figure 2: Examples of process of generating a new question with its explanation.

fore, affix/tense and preposition questions necessitate different focal points for extracting patterns and generating informative explanations.

**Affix/Tense Questions**   Affix/tense questions necessitate ESL learners to identify and analyze a specific context referred to as "hint words," which serve to modify or be modified by the word in the blank to answer the question accurately. To capture the patterns inherent in these questions, we focus on the relationship between the hint words and the answer option.

To extract the pattern from each expert-designed question, we begin by inserting the answer option into the sentence, resulting in a completed sentence denoted as $sent^{ans}$. Next, we extract the hint words from the expert-designed explanation, and we mark their corresponding positions in $sent^{ans}$ (see (a) in Figure 2). Subsequently, we employ dependency parsing on $sent^{ans}$ to generate its dependency tree. Given that the hint words and the answer option play crucial roles in the question, we extract a sub-tree from the dependency tree that encompasses all the hint words and the answer node. This sub-tree serves as the pattern for the question and is denoted as $pattern$ (see (b) in Figure 2, the pattern could be summarized as "A noun works as an object that is modified by an article and adjective.").

After obtaining the pattern for a specific question, we utilize it to generate new questions. We parse all sentences, denoted as $[\tilde{sent}_1, ..., \tilde{sent}_m]$, from publicly available news corpus to acquire their respective parsing trees, denoted as $[\tilde{tree}_1, ..., \tilde{tree}_m]$. We use a news corpus because news is in formal writing and leads to fewer grammatical errors. If a parsing tree, $\tilde{tree}_i$, includes

the extracted pattern $pattern_j$, we consider the corresponding sentence, $\tilde{sent_i}$, as a suitable candidate for generating a new question that belongs to $pattern_j$. It is important to note that our focus lies in capturing the modification relationship between the hint words and the answer option (e.g., dependency relations), and their grammatical classes within the sentence (e.g., POS), rather than the specific words used in the question generation process (see (c) in Figure 2).

To select distractors for the new question, we built candidate dictionaries for affix and verb-tense questions, respectively. Distractor options are selected from the corresponding dictionary. For example, if an affix question has the answer option "contractor", the distractor candidates could be in ["contractual", "contraction", "contracted", "contractable"]. Similarly, distractor options for verb-tense questions are also selected from another predefined dictionary.

Finally, we design templates for specific types of questions to present all the necessary information for answering the question, including $pattern$ and options (see (d) in Figure 2). To improve fluency and diversity, we employ LLM to paraphrase the template-based explanation. Details on the implementation can be found in Appendix B.

**Prep. Questions** Preposition questions require a comprehensive understanding of sentence meaning and the specific senses associated with the preposition options. Consequently, the pattern for a preposition question should incorporate the answer option along with its corresponding sense within the given sentence. To achieve this, we employed a preposition sense disambiguation (PSD) model to determine the sense of the answer option within a particular sentence, denoted as $sent^{ans}$.

Subsequently, we consider the answer option together with its identified sense as the pattern, denoted as $pattern$. We then apply PSD to sentences extracted from a publicly available news corpus. If a sentence, denoted as $\tilde{sent_i}$, contains the pattern $pattern_j$, it is considered a viable candidate for generating a new preposition question.

When selecting distractor options for preposition questions, a straightforward approach would involve randomly choosing prepositions from a pool of available options. However, this method may yield simple questions that are easy to answer. Such simplistic questions fail to effectively gauge the language proficiency of ESL learners or aid in

language learning (ALTE, 2011). As highlighted by Srikumar and Roth (2013), prepositions sharing the same semantic relation often appear in similar contexts. By utilizing prepositions with similar semantic roles as distractor options, we can enhance the difficulty level of preposition questions. To facilitate this, we construct a dictionary to cluster prepositions based on their semantic roles, which aids in the selection of appropriate distractor options.

Finally, similar to the approach described in Section 3.2.1, we design a template to generate initial explanations, which are then refined by employing an LLM to enhance their fluency and diversity. Details on the implementation can be found in Appendix B.

### 3.4 Dataset Analysis

To validate the quality and suitability of our created dataset for training models in the ClozEx task, we conducted a thorough manual quality assessment. As outlined in Section 3.1, the evaluation focused on two aspects: fluency and validity.

For the fluency assessment, we enlisted the expertise of two native English speakers from the university with which the authors are associated. These experts independently evaluated 100 randomly selected instances from our dataset using a 5-point Likert scale (1 denotes the worst and 5 denotes the best), solely considering the fluency of the generated explanations and disregarding their validity. To evaluate the validity aspect, we recruited four advanced ESL learners [3] from the university with which the authors are associated, because these learners possess a strong understanding of textbook grammar (Glisan and Drescher, 1993). Similarly, these annotators used a 5-point Likert scale to assess the validity of 100 instances. To ensure the independence between fluency and validity, we selected fluent instances in advance for the validity estimation. The validity assessment aimed to determine whether the explanations provided the necessary information to answer the corresponding question. Further details regarding the estimation process can be found in Appendix C.

To ensure robustness, each instance underwent double annotation for both fluency and validity. We performed the Pearson correlation test to assess the inter-annotator agreement between the different

---

[3]They hold public English test certificates to indicate they have a CEFR A2 level or higher.

| | IAA | | Estimation Score | | |
|---|---|---|---|---|---|
| | Pear. | $p$-value | Avg. | Med. | Var. |
| Flu. | 0.82 | <0.001 | 4.29 | 4.00 | 0.52 |
| Val. | 0.77 | <0.001 | 4.51 | 4.50 | 0.45 |

Table 2: Inter-annotator agreement and manual estimation result. **Pear.** denotes Pearson's correlation coefficient. **Avg.**, **Med.**, and **Var.** indicate the average, median, and variance of scores, respectively. **Flu.** and **Val.** represent fluency and validity.

| | #(Q, E) | Q avg. len. | E avg. len. |
|---|---|---|---|
| Train | 102,930 | 28.99 | 58.53 |
| Dev. | 22,056 | 29.00 | 58.69 |
| Test | 22,057 | 28.95 | 58.47 |

Table 3: Statistics of our dataset. **#(Q, E)** represents number of (question, explanation) pairs. **Q avg. len.** and **E avg. len.** denote average lengths of questions and explanations (number of words), respectively.

annotators. Result of inter-annotator agreement and manual estimation are shown in Table 2. The high correlation coefficients indicate a strong agreement among the annotators, underscoring the reliability of our manual estimation. The scores for both fluency and validity exhibited high median values and low variance. These findings confirm the high quality of our dataset and support its publication as a reliable resource for the ClozEx task.

For a comprehensive understanding of our dataset, Table 3 presents a statistical analysis, providing relevant insights into its characteristics.

## 4 Experiment

To address the ClozEx task, we conducted an investigation into baseline models under various scenarios and architectures. To evaluate the performance of these baseline models, we conducted thorough assessments using development and test data from our dataset, encompassing both manual and automatic evaluation metrics.

### 4.1 Experiment Setup

**Baseline Models** As a generation task, we employed encoder-decoder and decoder-only models for fine-tuning. In the case of the encoder-decoder models, we performed fine-tuning on BART (Lewis et al., 2020) and T5 (Raffel et al., 2020) architectures. For fine-tuning, we tailored cloze questions as input for the encoder-decoder models in the format of "$\{sent\}$[OPT]$\{opt_1\}$[OPT]...$\{opt_4\}$,"

where "[OPT]" is a special token that is used for concatenation among sentence and options. The output of the encoder-decoder models is the corresponding explanation. We explored different model sizes, including base and large, to assess their performance in the ClozEx task.

On the other hand, in the case of the decoder-only models for fine-tuning, we selected GPT2 and GPT2-medium (Radford et al., 2019). For decoder-only models, the input is a question with an explanation that is connected with a prompt. We then fine-tuned models with such input instances.

Because LLMs have shown remarkable performance across diverse tasks in zero-shot scenarios (Kojima et al., 2022), to explore the potential of LLMs in solving the ClozEx task without the need for additional training data, we employed LLMs of different sizes and structures to generate explanations without fine-tuning. We employed GPT2-large, GPT2-XL, GPT3.5-turbo [4], and LLaMa-7B (Touvron et al., 2023) to generate explanations in the zero-shot scenario. The prompts used for the LLMs can be found in Appendix D.

**Evaluation Metrics** We engaged human annotators to estimate the fluency and validity of the generated explanation, following the same estimation process as described in Section 3.4. We randomly selected 100 samples of generated explanations from each model to be estimated. All instances were estimated without reference explanations, ensuring a reference-free evaluation.

To complement the manual annotation, which can be time-consuming and less generalizable, we also employed automatic metrics to assess the generated explanations. For reference-based metrics, we used BLEU-4 (Papineni et al., 2002) from the Huggingface Evaluate library [5] to measure the similarity between the generated explanations and the reference labels. According to Wang et al. (2023a), LLMs such as GPT3.5-turbo can evaluate the quality of generated text and exhibit a moderate correlation with human annotators. Therefore, we utilized GPT3.5-turbo as a reference-free metric to evaluate the fluency and validity of the generated explanations. The reliability of GPT evaluators will be discussed in Section 5.2. Samples used for the GPT evaluator are the same as manual estimation. All metrics except BLEU are based on the Likert 5-point scale. Prompts for the GPT evaluator can

---

[4] https://platform.openai.com/docs/models
[5] https://huggingface.co/docs/evaluate/index

be found in Appendix D.

## 4.2 Result

The evaluation results are presented in Table 4. With regard to the manual metrics, the encoder-decoder models generally exhibited the ability to generate fluent and valid explanations, except for T5-base. BART-large achieved the highest level of validity performance. By contrast, the decoder-only models based on GPT-2 produced acceptably fluent texts but did not effectively explain the questions. Across all the fine-tuned models, the size of the model did not have a substantial impact on performance, except for T5 base and large, where it hindered the generation of more valid explanations. Although LLMs are capable of generating text with acceptable fluency thanks to the large amount of pre-training data, they received low evaluations in terms of producing valid explanations. This highlights the ongoing challenge of using LLMs to generate cloze question explanations for LA, without mentioning the generation of a dataset specifically tailored for the ClozEx task. A detailed discussion regarding the performance of LLMs is included in Section 5.1.

With regard to the automatic metrics, the BLEU score exhibited a strong correlation with manual fluency and validity scores when evaluating models fine-tuned with our training data. However, because the LLMs did not learn the distribution from our training data, the generated text varied from the reference. Because a good explanation for a cloze question is not unique, reference-based metrics should focus on evaluating models trained with our data. In this regard, BART-large achieved the best performance once again.

The GPT evaluator demonstrated stability in terms of fluency. These GPT-Fluency scores showed a positive correlation with manual fluency scores. However, in terms of validity, the GPT evaluator was less consistent, assigning varying scores to models that received similar validity scores from human annotators (such as BART-base, BART-large, and T5-large). Notably, LLM-GPT3.5-turbo was highlighted, because the GPT evaluator exhibited more leniency toward it than human annotators.

Finally, although these automatic scores showed some correlation with human evaluation, they were calculated under the macro average. To determine the reliability of these automatic metrics in the ClozEx task, we will discuss the micro-averaged Pearson correlation coefficient between manual and automatic scores in Section 5.2.

## 5 Discussion

### 5.1 Do LLMs Explain Cloze Questions Well?

Given the remarkable performance of LLMs across various tasks without fine-tuning (Liu et al., 2023), there is a reasonable expectation that they would excel in generating high-quality explanations for cloze questions. However, our experimental findings indicate that no LLM achieved an acceptable validity score in manual evaluation. Upon analyzing the explanations generated by GPT3.5-turbo, we identified two critical shortcomings of LLMs in effectively explaining cloze questions.

Firstly, LLMs exhibit a tendency to generate factual errors, thereby failing to ensure the accuracy of the generated texts. This deficiency is exemplified in **LLM-GPT3.5-turbo Question 1** Appendix E, where an evident error is observed in the verb tense following the word "did not," a discrepancy that can have detrimental consequences in the context of LA.

Secondly, LLMs have the propensity to produce explanations that lack meaningful and informative content, failing to provide the necessary knowledge required for comprehending the reasons behind the answer options. As illustrated by **LLM-GPT3.5-turbo Question 2** in Appendix E, such explanations leave ESL learners unaware of why the given answer option is necessary, while also failing to elucidate the distinctions among the options resulting from affixes. Furthermore, these explanations may even present incorrect answers and flawed analyses, further diminishing their utility.

### 5.2 Are Automatic Metrics Reliable in ClozEx?

The evaluation of automatic metrics, specifically BLEU and GPT-Fluency scores, aligns with the trends observed in manual evaluation scores (Section 4.2). To ascertain the reliability of these metrics in reflecting the quality of generated explanations, we computed the micro-averaged Pearson correlation coefficient between manual and automatic evaluation scores.

As shown in Table 5, the BLEU score is largely independent of the manual fluency score. However, when excluding explanations generated by LLMs, the BLEU score exhibits a moderate cor-

| | Manual | | Automatic | | |
|---|---|---|---|---|---|
| | Fluency | Validity | BLEU | GPT-Fluency | GPT-Validity |
| BART-base | 4.13 | 4.38 | 25.64 / 25.53 | 4.88 | 3.75 |
| BART-large | 4.11 | **4.43** | **27.33 / 27.01** | 4.84 | 2.90 |
| T5-base | 2.03 | 1.52 | 7.62 / 7.59 | 2.53 | 1.32 |
| T5-large | 3.99 | 4.26 | 22.70 / 22.68 | **4.95** | 2.31 |
| GPT2 | 3.87 | 2.78 | 15.40 / 15.41 | 4.03 | 1.77 |
| GPT2-medium | 3.91 | 1.85 | 16.85 / 16.84 | 4.16 | 2.03 |
| LLM-GPT2-large | 3.97 | 1.73 | 0.50 / 0.51 | 3.94 | 1.58 |
| LLM-GPT2-XL | 3.97 | 1.70 | 0.60 / 0.60 | 4.00 | 1.58 |
| LLM-GPT3.5-turbo | **4.53** | 2.70 | 1.39 / 1.34 | 4.93 | **4.87** |
| LLM-LLaMa-7B | 3.81 | 1.78 | 1.06 / 1.08 | 3.81 | 1.44 |

Table 4: Performance of baseline models. BLEU scores are based on dev. and test sets, respectively. In evaluation metrics, GPT-Fluency and GPT-Validity indicate fluency and validity estimation, respectively, using GPT3.5-turbo. Prefix LLM- denotes LLM-generated explanations. Except for BLEU, all scores are ranged in [1, 5].

| | Man. Fluency | Man. Validity |
|---|---|---|
| BLEU | 0.04 / 0.17 | 0.08 / 0.11 |
| - w/o LLMs | 0.39 / 0.43 | 0.44 / 0.47 |
| GPT-Fluency | 0.57 / 0.61 | — |
| GPT-Validity | — | −0.03 / 0.05 |

Table 5: Pearson correlation coefficient between manual and automatic evaluation scores. The automatic scores yielded two correlated coefficients because each instance is assessed by two annotators.

relation with the manual fluency score. The validity correlation reported a similar tendency. As a reference-based metric, BLEU demonstrates limitations in recognizing explanations with different styles from our dataset, implying that a low BLEU score does not necessarily indicate a poor explanation. However, due to the high quality of our dataset, an explanation with a high BLEU score can generally be considered good.

As a reference-free metric, GPT-Fluency exhibits a strong correlation with manual fluency scores, even when considering LLM explanations. Unlike the correlation observed between GPT-Fluency and Manual Fluency, GPT-Validity fails to effectively reflect the manual validity score. Furthermore, for explanations generated by LLM-GPT3.5-turbo, as mentioned in Section 5.1, GPT-Validity tends to assign higher scores. In light of these findings, when a reference-free evaluation is conducted, it is acceptable to employ LLMs such as GPT3.5-turbo to assess fluency in the ClozEx task. However, using LLMs to evaluate validity is not recommended.

# 6   Conclusion

This paper introduced a novel task, ClozEx, aimed at generating fluent and valid explanations for English cloze questions to support ESL learning.

We curated a comprehensive dataset comprising more than 140k instances of cloze questions paired with explanations. The dataset is created by a pattern-based method. Patterns extracted from expert-designed cloze questions and explanations ensure the quality of generated questions and explanations. Expert evaluations confirmed the high quality and suitability of our dataset for the ClozEx task.

To address this task, we fine-tuned various models, including encoder-decoder and decoder-only architectures, to generate explanations for the provided questions. Additionally, we investigated the potential of LLMs to produce explanations in a zero-shot scenario. The experimental results highlighted the capability of the encoder-decoder models to generate high-quality explanations. However, although LLMs excelled in generating fluent texts, they struggled to produce valid explanations. Thus, we analyzed the limitations of LLMs in generating satisfactory explanations without fine-tuning, shedding light on the challenges they face in this context.

Additionally, we explored the correlation between manual and automatic evaluation metrics, discovering that automatic metrics exhibited some degree of reliability for the ClozEx task.

## Limitations

One limitation of our study on the `ClozEx` task, designed to support ESL learning, is that our experiment did not investigate its effectiveness in improving language proficiency. Although the expert estimation of our dataset yielded positive results, it serves only as indirect evidence that the explanations contained within can aid ESL learning. To obtain direct evidence, further experiments are required. For instance, conducting a study where English proficiency of ESL learners is assessed before and after exposure to a batch of questions and explanations from our dataset would allow us to observe whether these materials contribute to proficiency enhancement.

Another limitation of our study pertains to the question types included in the dataset. Initially, we constructed the dataset by employing pattern extraction methods that focused on three specific question types. However, it is important to note that language assessment encompasses a wide range of question types. For instance, there are questions that require learners to identify the meanings of content words or assess the usage of pronouns, conjunctions, and other linguistic elements. The pattern extraction methods utilized in our study were tailored to address specific question types, which may limit the coverage of our dataset. To expand the scope of our dataset, future efforts would entail devising new methods to extract patterns from these specific question types.

The third limitation pertains to the automatic evaluation metrics employed. Although we observed a positive correlation between BLEU and GPT-Fluency scores and manual evaluation scores, certain issues arise when these metrics are utilized. BLEU, being reference-based, encounters difficulties when confronted with situations where good explanations are not unique. Although reference-free metrics like LLMs, such as GPT-Fluency, offer an alternative, their reliability is not always guaranteed (Wang et al., 2023b). Additionally, models such as GPT3.5-turbo, which served as the backbone for GPT evaluators in our study, are not open-source, posing potential obstacles for future research endeavors.

Furthermore, despite demonstrating proficiency in paraphrasing explanations during the dataset creation phase, LLMs proved inadequate in generating explanations for cloze questions without any prior information. These observations underscore the limitations of LLMs when it comes to effectively elucidating cloze questions. Overcoming these challenges will necessitate the exploration of novel methodologies and strategies, such as incorporating external grammatical knowledge, to enhance the ability of LLMs to generate explanations that are precise, informative, and contextually appropriate.

## Acknowledgements

This work was supported by JST, the establishment of university fellowships towards the creation of science technology innovation, Grant Number JP-MJFS2139.

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

## A   Examples of Questions in Dataset

## B   Details of Dataset Creation

Patterns for affix/tense questions were extracted from a published TOEIC practice book (Kato, 2017). A total of 231 patterns were extracted from 432 affix questions, while 99 patterns were extracted from 219 tense questions. For preposition questions, we focused on 34 prepositions used in the PSD dataset (Gong et al., 2018) as question patterns.

To generate new questions and explanations, we selected the ag_news (Zhang et al., 2015), cc_news (Hamborg et al., 2017), and multi_news (Fabbri et al., 2019) corpora from the public news corpus.

In the process of creating new preposition questions, we employed BERT-PSD [6] to identify the pattern present in each given sentence. Although BERT-PSD is a state-of-the-art model in the PSD task, it achieved an accuracy of only 90.84%, leading to potential noise in the dataset. To address this, we set a threshold of 0.8 for the model's prediction confidence. If the model predicted the pattern of a sentence with a confidence equal to or higher than 0.8, we retained the sentence along with its pattern for producing new questions and explanations. Otherwise, the sentence was discarded. With

---

[6] https://github.com/dirkneuhaeuser/preposition-sense-disambiguation

---

**Affix Question**:

As expected, the infectious period had a clear ____ relationship with mean offspring number.

(A) positive (B) positively (C) positives (D) positivity

**Explanation**:

For this cloze question, you need to choose an adjective that modifies the word "relationship." Option (A) "positive" is an adjective, while options (B), (C), and (D) are not adjectives. Therefore, the correct answer is option (A), "positive."

**Verb Tense Question**:

The couple were ____ Saturday on disorderly conduct charges by officers investigating a family dispute at their home in New Canaan, Conn.

(A) arresting (B) arrested (C) arrest (D) arrests

**Explanation**:

The blank in the cloze question requires a past participle verb, as indicated by the passive auxiliary "were" and the passive nominal subject "couple." The options given are: (A) arresting (gerund or present participle), (B) arrested (past participle) (C) arrest (base form), and (D) arrests (3rd person singular present). Based on this information, the correct answer is option (B), "arrested."

**Preposition Question**:

The work was done ____ the Rehabilitation Institute of Chicago under an $8-million grant from the Army.

(A) at (B) down (C) round (D) of

**Explanation**:

Based on the context of the sentence, option (A) "at" is the appropriate choice for the cloze question. "At" indicates a specific location or arrival at a particular place or position, such as "he is at the store."

Table 6: Examples of questions with different types.

this threshold, the prediction accuracy improved to 97.78%.

For creating distractor options in affix questions, we prepared a distractor candidate dictionary in advance. We collected words from an English dictionary website[7] that share the same root but have different prefixes or suffixes. A similar process was followed for tense questions, where the distractor candidate dictionary focused specifically on verbs and their various tense forms. In the case of preposition questions, the distractor candidate dictionary was created based on preposition semantic relations (Srikumar and Roth, 2013). Prepositions that share the same semantic relations are considered as distractor options for each other.

To avoid ambiguous questions that have multiple correct answers, we utilized a GPT2-based LM scorer [8]. If a distractor option obtained a higher LM score than the answer, as determined by the scorer, the option was discarded. The templates used for generating initial explanations for questions are shown in Table 7. These initial explanations were further paraphrased using GPT3.5-turbo. The prompt for paraphrasing is provided in Appendix D.

## C    Details of Manual Quality Estimation

Human evaluators were tasked with rating the quality of generated explanations from each method in terms of fluency and validity using a 1-5 scale. The following instructions and criteria were provided to guide their ratings:

**Fluency.**    You are given instances of English fill-in-the-blank questions with corresponding explanations. Your task is to estimate whether the explanation is fluent in English. For a batch, you need to estimate 45 instances. You need to estimate the fluency using a 5-scale metric to score the explanation, and you do not need to identify whether the explanation explains the question correctly, please just focus on its fluency. The ratings are as follows:

- 1=Bad: The explanation was unreadable.

- 2=Unacceptable: The explanation was disfluent.

---

[7] https://www.vocabulary.com/
[8] https://github.com/simonepri/lm-scorer

- 3=Borderline: The explanation fell between unacceptable and acceptable fluency.

- 4=Acceptable: The explanation was clear and understandable, but with room for improvement.

- 5=Good: The explanation was fluent and easy to understand.

**Validity.**    You are given instances of English fill-in-the-blank questions with corresponding explanations. Your task is to estimate whether the explanation explains the question well. For a batch, you need to estimate 45 instances. You need to estimate the validity using a 5-scale metric to score the explanation. The ratings are as follows:

- 1=Bad: The explanation included factual errors or was unrelated to the question.

- 2=Unacceptable: The explanation was related to the question but provided knowledge that did not contribute to answering it.

- 3=Borderline: The explanation fell between unacceptable and acceptable validity.

- 4=Acceptable:  The explanation provided some necessary knowledge for answering the question, but there were still some missing elements.

- 5=Good: The explanation provided sufficient language knowledge to answer the question.

The annotation was approved by the ethical committee in the authors' university prior to conducting this research (approval number: H21-041). All annotators were paid about $13.5 for every 45 instances (which takes about 1 hour), while the minimum wage locally is about $7.5.

## D    Used Prompts

The prompt we used to paraphrase initial explanations with GPT3.5-turbo (the parameter of OpenAI API) is:

messages=[

{'role': 'system', 'content': 'You are an English teacher.'},

| | |
|---|---|
| Affix/tense | The word in the blank should be $\{RELATION\_TO\_PARENT\_OF\_ANSWER\}$ of "$\{PARENT\_OF\_ANSWER\}$". "$\{CHILD_i\_OF\_ANSWER\}$" is $\{RELATION\_TO\_CHILD\_OF\_ANSWER\}$ of the blank. Thus, a $\{POS\_OF\_ANSWER\}$ is required. $\{OPTION_i\}$ is a $\{POS\_OF\_OPTION_i\}$. Therefore, the correct answer is $\{ANSWER\}$. |
| Prep. | According the meaning of this sentence the option $\{ANSWER\}$ is suitable, which means "$\{SENSE\_OF\_ANSWER\}$". |

Table 7: Templates used for generate initial explanations.

{'role': 'user', 'content': f'Paraphrase the following explanation of a cloze question within 128 words: $\{exp\}$'}

]

The prompt we used in training/inference with LLM-GPT2 family and LLM-LLaMa-7B is:

Question: $\{sent\}$\nOptions: (A) $\{opt_1\}$ (B) $\{opt_2\}$ (C) $\{opt_3\}$ (D) $\{opt_4\}$\nExplanation:$\{exp\}$

where $exp$ is set to empty in the inference phase.

The prompt we used in inference with GPT3.5-turbo is:

messages=[

{'role': 'system', 'content': 'You are an English teacher.'},

{'role': 'user', 'content': f'Generate an explanation of the following cloze question: $\{sent\}$\nOptions:(A) $\{opt_1\}$ (B) $\{opt_2\}$ (C) $\{opt_3\}$ (D) $\{opt_4\}$'}

]

For GPT-evaluators, the prompt we used in GPT-Fluency is:

messages=[

{'role': 'system', 'content': 'You are an English teacher.'},

{'role': 'user', 'content': f'Estimate whether the given text is fluent.\nHere is the score definition:\n1=Bad: The explanation was unreadable.\n2=Unacceptable: The explanation was disfluent. \n3=Borderline: The explanation fell between unacceptable and acceptable fluency. \n4=Acceptable: The explanation was clear and understandable, but with room for improvement.\n5=Good: The explanation was fluent and easy to understand.\nThe input is: $\{exp\}$'}

]

In GPT-Validity, the prompt is:

messages=[

{'role': 'system', 'content': 'You are a linguistics expert.'},

{'role': 'user', 'content': f'Estimate whether the given explanation could explain the cloze question well.\nHere is the score definition:\n1=Bad: The explanation included factual errors or was unrelated to the question.\n2=Unacceptable: The explanation was related to the question but provided knowledge that did not contribute to answering it.\n3=Borderline: The explanation fell between unacceptable and acceptable validity.\n4=Acceptable: The explanation provided some necessary knowledge for answering the question, but there were still some missing elements.\n5=Good: The explanation provided sufficient language knowledge to answer the question.\nThe input question is: $\{q\}$\nThe explanation is: $\{exp\}$'}

]

## E  Examples of Generated Explanations

**BART**

**Question:** The repayment, which settles a suit brought under an international legal tribunal, is ____ from the tens of billions of dollars in frozen foreign accounts that Iran can now access after the end of nuclear sanctions.

(A) separates (B) separability (C) separateness (D) separate

**Explanation (BART-base):** The blank in the cloze question is filled by the copula "is". This means that an adjective is needed to complete the sentence. The options given include a verb in the third person singular present tense, a singular or mass noun, an adjective, and another adjective. The correct answer is the adjective "separate".

**Explanation (BART-large):** The blank in the cloze question is filled by the copula "is". This means that an adjective is needed to complete the sentence. The options given are: (A) separates, which is a verb in the 3rd person singular present tense; (B) separability, a singular or mass noun; (C) separateness, also a noun in the same category; and (D) separate, an adjective. Based on this information, the correct answer is option (D), which is "separate".

**T5**

**Question:** South Korean shares slid on Friday, with foreign investors _____ their biggest daily volume in nearly 20 months, hurt after aggressive easing from the Bank of Japan sent the yen reeling, as well as by the tension over North Korea.

(A) selling (B) undersells (C) resell (D) sell

**Explanation (T5-base):** The stock market slid on Friday, with investors

_______________________________

**Explanation (T5-large):** The correct answer for the cloze question is (A) selling, which is a verb in the form of a gerund or present participle. This is indicated by the fact that the word "selling" can be used as a verb in this context.

**GPT2**

**Question:** The father of the former Meghan Markle says he wishes he could have walked her _____ the aisle during her wedding to Prince Harry.

(A) with (B) down (C) as (D) for

**Explanation (GPT2):** Based on the definition provided, option (b) "down" is the appropriate choice for the blank in this sentence. This word indicates movement towards a lower position or level, as in the example "he descended the hill." This word can also be used to describe the action of

**Explanation (GPT2-medium):** The correct answer for the cloze question is option (b) down, which refers to the action of moving towards a lower point or level, such as when someone goes down a hill. This is based on the sentence's meaning. For example, "he

**LLMs (except GPT3.5-turbo)**

**Question:** The repayment, which settles a suit brought under an international legal tribunal, is _____ from the tens of billions of dollars in frozen foreign accounts that Iran can now access after the end of nuclear sanctions.

(A) separates (B) separability (C) separateness (D) separate

**Explanation (LLM-GPT2-large):** The first option is to separate the accounts from the rest of the assets. This would allow the U.S. to keep the money in the accounts, but not the assets. The second option is to separate the accounts from the rest of the assets. This would allow the U

**Explanation (LLM-GPT2-XL):** Separation is the most common option. Separation is the most common option. Separation is the most common option. Separation is the most common option. Separation is the most common option. Separation is the most common option. Separation is the most common option. Separation is the most common option. Separation is the most common option. Separation is the most common option.

**Explanation (LLM-LLaMa-7B):** The repayment, which settles a suit brought under an international legal tribunal, is from the tens of billions of dollars in frozen foreign accounts that Iran can now access after the end of nuclear sanctions.

Question: The repayment, which settles a suit brought under an international legal tribunal, is from the tens of billions of dollars in frozen foreign accounts that Iran can now access after the end of nuclear sanctions

**LLM-GPT3.5-turbo**

**Question 1:** Isn't it just that it didn't ____ to sell the sets it had produced?

(A) manage (B) manages (C) managing (D) managed

**Explanation 1:** The correct answer is (D) managed. The sentence is in the past tense and "did not" is the negative form of the auxiliary verb "did". Therefore, the next verb has to be in the past tense as well. The verb "managed" is the correct form in the past tense, meaning "succeeded in doing something despite difficulties or obstacles". Example: "The company didn't manage to sell all of its products before the end of the year."

**Question 2:** The innards of the machine were complex but ____.

(A) manageable (B) manageably (C) unmanageably (D) mismanage

**Explanation 2:** The correct option is (C) unmanageably. The word "innards" means the internal parts of a machine. The sentence suggests that the parts of the machine were complex and difficult to manage. The word "manageably" (option B) means something that is able to be managed easily which contradicts the sentence. The word "mismanage" (option D) means to handle or operate wrongly or inefficiently, which does not fit in the sentence. Therefore, the correct answer is (C) unmanageably, meaning that the innards of the machine were difficult to manage.