# OpenReview forum: "ClozEx: A Task toward Generation of English Cloze Explanation"
_EMNLP/2023/Conference — EMNLP 2023 Findings_

### Official Review · Reviewer_zBsN · 2023-07-29

**Soundness:** 4

**Excitement:**

3: Ambivalent: It has merits (e.g., it reports state-of-the-art results, the idea is nice), but there are key weaknesses (e.g., it describes incremental work), and it can significantly benefit from another round of revision. However, I won't object to accepting it if my co-reviewers champion it.

**Justification For Ethical Concerns:**

Original review:
No mention of appropriate review board
No mention of if annotators were fairly compensated and how the compensation was determined to be fair
Please see “For papers presenting new datasets” in
https://2023.emnlp.org/ethics/review-questions/

Edit:
The authors have clarified that the annotation work was approved by the ethical committee and that annotators were paid above minimum wage to perform the annotations

**Paper Topic And Main Contributions:**

The authors propose a new task: ClozEx which focuses on generating explanations for cloze question in language assessment
For this task, they present a new dataset created from templates/pattern applied to a corpus of news text
The authors try training and prompting a number of large language models (LLM) to be used as baseline for the proposed task and dataset.
They access the questions created for the dataset and the questions generated by the language models using human annotations of Fluency (how easy the explanation is to comprehend) and Validity (is the explanation valid) on the Likert scale. Additionally, author evaluate the LLM generated questions using some automatic evaluation methods
The fine tuned encoder-decoder LLM methods seems to outperform decoder only method and prompting methods with the BART-large baseline archives the highest Validity score
Finally, the authors compare the correlation of human and automatic evaluation scores
GPT-Fluency and annotator Fluency are highly correlated. However, none of the automated evaluation methods were not highly correlated with annotator Validity.

**Questions For The Authors:**

1. Are the same annotators used for evaluating the LLM generated explanations at the dataset explanations? (And is there a time gap between those 2 annotations or were they annotated at the same time)
2. Also, were the annotators paid for the annotators as part of their salary, paid for completing the annotations or volunteers?
(You may want to add an ethics statement discussing pay and other ethics consideration of the annotations)

**Reasons To Accept:**

1. The task may and be useful for developing language learning systems
2. I think the insights into the relationship between automated evaluation methods and human evaluation methods is useful

**Reasons To Reject:**

1. The 6 annotators are recruited from the author’s university. This may introduce bias towards questions in the dataset and makes the annotations difficult to reproduce. It is unclear if they knew the origin/context explanations they were annotating. It is also I think it would be performing annotations on a platform like M-Turk. I would also consider including the full instructions provided to annotators in Appendix C.

Edit:
The authors have explained why they did not use a crowd source annotation platform like M-Turk and have provided the instructions shown to the annotators. However, not all annotations were performed at the same time (annotations were performed with a 2 week gap). It is possible that this could produce some bias in the annotations. This gap should be noted in the Limitations for this work.

**Reproducibility:**

4: Could mostly reproduce the results, but there may be some variation because of sample variance or minor variations in their interpretation of the protocol or method.

**Reviewer Confidence:**

4: Quite sure. I tried to check the important points carefully. It's unlikely, though conceivable, that I missed something that should affect my ratings.

**Typos Grammar Style And Presentation Improvements:**

Line 538: “no LLM achieved an acceptable validity score in manual evaluation”. I think you mean “no LLM achieved an acceptable validity score in manual evaluation without fine tuning” as it seems like BART-large explanations perform about the same at the original dataset explanations

---

> ### Author Rebuttal · Authors · 2023-08-28
>
> Thank you for your feedback and the concern about the ethics risk.
>
> In the two phases of annotation, we hired the same six annotators, but there was an obligatory time gap between the two annotations. There were about two weeks between two annotations. During that period we needed to train baseline models and obtain results for the next annotation process.
>
> Because all instances in annotation were automatically generated, and we did not leak them to annotators, there was no chance for the annotators to see the instances before. Instruction is listed at the end of our response.
>
> We have considered using MTurk to perform the annotation, but we could not identify the English level of a learner. Hiring English learners from the university could solve it --- we checked annotators' English test certificates (e.g., IELTS, TOEFL, TOEIC) physically to make sure they have CEFR A2 level or higher.
>
> The annotation was approved by the ethical committee in the authors' university prior to conducting this research. All annotators were paid about 13.5 USD for every 45 instances (takes about 1 hour), and the minimum wage locally is about 7.5 USD.
>
> We appreciate your thoughtful consideration, and we will add the above points to our paper.
>
> -----
>
> Instruction of fluency:
> You are given instances of English fill-in-the-blank questions with corresponding explanations (See example in Table 1). Your task is to estimate whether the explanation is fluent in English. For a batch, you need to estimate 45 instances. The reward for each batch is an Amazon gift card of \$13.5.
>
> Table 1(Omitted)
>
> You need to estimate the fluency using a 5-scale metric to score the explanation, and you don’t need to identify whether the explanation explains the question correctly, please just focus on its fluency.
>
> The 5 (the best) represents the explanation was fluent and easy to understand.
>
> The 4 represent the explanation was clear and understandable, but with room for improvement.
>
> The 3 represent the explanation that falls between unacceptable and acceptable fluency.
>
> The 2 represent the explanation was disfluent.
>
> The 1 (the worst) represents the explanation was unreadable.
>
> -----
>
> Instruction of validity:
> You are given instances of English fill-in-the-blank questions with corresponding explanations (See example in Table 2). Your task is to estimate whether the explanation can explain the question well. For a batch, you need to estimate 45 instances. The reward for each batch is an Amazon gift card of $13.5.
>
> Table 2 (Omitted)
>
> You need to estimate the validity using a 5-scale metric to score the explanation.
>
> The 5 (the best) represents the explanation that provided sufficient language knowledge to answer the question.
>
> The 4 represents the explanation that provided some necessary knowledge for answering the question, but there were still some missing elements.
>
> The 3 represents the explanation that falls between unacceptable and acceptable validity.
>
> The 2 represents the explanation that was related to the question but provided knowledge that did not contribute to answering it.
>
> The 1 (the worst) represents the explanation that included factual errors or was unrelated to the question.

---

### Official Review · Reviewer_xPm4 · 2023-08-02

**Soundness:** 3

**Excitement:**

2: Mediocre: This paper makes marginal contributions (vs non-contemporaneous work), so I would rather not see it in the conference.

**Paper Topic And Main Contributions:**

The paper propose a contribution about English Cloze Explanation. The authors propose a method to build explanation dataset and conduct some experiments showing that the explanation generation task is challenge.

**Reasons To Accept:**

1. The task about Expalanation generation is interensting.
2. The paper propose a dataset about explanation generation.

**Reasons To Reject:**

At first, the experimental part of this paper is extremely imprecise:
1. To my knowledge, explanation generation task is special so the BLEU metric is unsuitable. For the exmaple in figure 1, the explanation "Based on the context of the sentence, option A, option B, option C, so the answer is " can obtain high BLEU score, but this explanation is unhelpful. For this reason, all the BLEU in Table 4 is meaningless. For example, we can see T5-base only obtain 7.59 of BLEU score, but BART-base can obtain 25.53.
2. In lines 439-449, the authors claim that they do not fine-tune the LLMs and compare the results with other baselines. I think it is so unfair comparisions, which makes me very curious about the actual performance of LLMs. So I test some cases about cloze explanation in ChatGPT. In fact, ChatGPT can provide good explanations for most cases. I think show a so low BLEU score in Table 4 for LLMs but lack details is a serious problem for this paper.
3. In Table 4, the authors use GPT-model as a metric to evaluate the models including many GPT-based models, so these two metrics are unconvincing.
So all the results in Table 4 are meaningless except human evaluation, but human evaluation only test 100 samples as describe in line 454.

Secondly, as shown in section 3.2, all the explanations and questions are generated by template, which are far away from real ESL education. One of the most prominent problems is the lack of difficulty. Although the paper provide a large-scale dataset including 140k samples, there is a high possibility that it contains a large number of similar samples. I think the authors should conduct some experiments  about this problem.

Thirdly, the paper constructs a dataset by constructing a corpus through the template and then polishing it with LLMs. We can see that the authors have been provided the pipeline to solve ClozEx dataset. It is meaningless to generate explanations by a end-to-end method for these data.

I think it should at least provide a dataset where all questions and explanations are written by humans to be meaningful for research.

Furthermore, the authors write so many text in section Limitations. Although the reviewing guidelines suggest me that I could not reject the paper base on the limitations. But I think I must put forward my point of view. If a method is not good, what researchers should do is not to use it and explore better methods. Rather than saying that my work uses this method, I admit that it is not good but it has nothing to do with my work. The method you use is also part of your work.

**Reproducibility:**

4: Could mostly reproduce the results, but there may be some variation because of sample variance or minor variations in their interpretation of the protocol or method.

**Reviewer Confidence:**

5: Positive that my evaluation is correct. I read the paper very carefully and I am very familiar with related work.

---

> ### Author Rebuttal · Authors · 2023-08-28
>
> Thank you for your careful feedback. Here are responses to your questions:
>
> 1. To Reasons To Reject: “To my knowledge...”.
>
>     Thank you for raising this point. We agree with you that while evaluating the model performance, manual evaluation is, of course, the most accurate metric. However, the manual evaluation is costly and lacks reproducibility, so, in addition to the manual evaluation,  we explored whether automatic metrics could represent the model performance as well.  It is not obvious to what extent BLEUs are reliable in this generation task, and we consider it significant and a contribution of this study to have clarified this point.
>     The result in Section 5.2 demonstrated that reference-based metric has indeed its own limitations and could only be performed in specific conditions.
>
>
> 2. To Reasons To Reject: “In lines 439-449...”.
>
>     Thank you for taking your time to test some cases for our task! Because our paper is going to propose a new task and dataset, under the condition of limited pages, we prefer leaving more space for introducing the task and dataset, as well as analysis of several baseline models to provide insights to future work.
>
>     Another reason that we did not fine-tune LLMs was because we wanted to investigate whether LLMs could tackle the ClozEx task well in a zero-shot scenario --- if LLMs could do so  without any external data, the ClozEx training set is meaningless (i.e., only the dev/test sets are useful). Therefore, LLMs in our experiment were not expected to be compared with fine-tuned models.
>
>     You mentioned that ChatGPT could produce acceptable explanations in your experiment. We are interested in understanding the discrepancy between your experiment and ours. However, because we are not familiar with your experiment setting, it is not easy to answer what caused the inconsistency. Could you kindly share your experimental settings (actual prompts) and examples (if not from our data)? Constructive discussions around such differences can lead to valuable insights. Here we would like to share some of our opinions (just FYI, our zero-shot prompts are included in Appendix D).
>
>     In our experiment,  GPT3.5-turbo did not perform as poorly as other LLMs and obtained valid scores near the borderline. Furthermore, as indicated in Section 5.1,  GPT3.5-turbo suffered from two typical problems. We attach some examples of questions at the end of this response. These questions are from the ClozEx dataset and cause poor explanations when using GPT3.5-turbo API (results are reproducible because we set Temperature=0).
>
>     As for low BLEU scores of LLMs, as we indicated in Section 5.2 and Limitations, reference-based metrics are not appropriate to evaluate explanations from zero-shot generation, which is why LLMs generally did not reach high BLEU scores. We listed the BLEU scores of LLMs to demonstrate it is not suitable in such cases.
>
>
> 3. To Reasons To Reject: “In Table 4, the authors use GPT-model...”.
>
>     Thank you for bringing our attention to the manual evaluation. As you pointed out, the number of manual evaluations was limited to 100 samples. However, we believe that the number was sufficient because inter-annotator agreements for fluency and validity were 0.82 and 0.77, respectively (statistically significant, Table 2).
>
>     We did not claim whether an automatic metric is convincing before calculating the correlation with the manual metric. To reduce the cost and improve the reproducibility of model evaluation, we explored whether automatic metrics are consistent with manual ones --- if automatic metrics can represent the model performance as well as manual metrics, future work will not have to compare models by manual annotation. In Section 5.2, the correlation indicated that not all automatic metrics are reliable in evaluating ClozEx methods, some are totally unusable and some can only work on limited conditions. We kindly ask the reviewer to note that only GPT3.5-turbo is using the same model with GPT-Fluency and GPT-Validity (although there may be overlap between the training dataset between GPT-based models).
>
> 4. To Reasons To Reject: “Secondly, as shown in section 3.2...”.
>
>     Our questions are generated by expert-designed questions, which cover various grammatical items with different difficulties. We extract patterns from such questions and use extracted patterns in turn to generate questions, so generated questions could be regarded as an expansion of expert-designed questions. Instances in our dataset are from fixed patterns, but question sentences and options are selected randomly in limited conditions from three large corpora, which could avoid the domain dependency. Nonetheless, thanks for suggesting an experiment for additional experiments and we would love to include statistics showing the variety of the created dataset. In the meantime, we would like the reviewers to download our data accompanied with this submission.
>
> 5. To reasons To Reject: “Thirdly, the paper constructs a dataset...”.
>
>     The pipeline of creating the dataset requires both expert-designed questions and explanations. It is hard to generate explanations using the pipeline by only providing questions. Therefore, the ClozEx task requires models to generate explanations with only questions as input.
>
> We are sorry for the misunderstanding due to our unclear writing. We would modify the draft by emphasizing the above items.
>
> -----
>
> Examples of questions leading to poor explanation (Using GPT3.5-turbo API; Temperature=0; prompt is indicated in Appendix D in our paper; verified on 24/08/2023.)
>
> Example 1:
> > Question: The protective netting in place to protect fans from falling concrete at Wrigley Field must be replaced with a permanent solution if the Chicago Cubs _______ to play at the stadium next spring.
> >
> > Options: (A) wanting  (B) want  (C) wanter  (D) wants
>
> Example 2:
> > Question: Isn't it just that it didn't _______ to sell the sets it had produced?
> >
> > Options: (A) manage (B) manages (C) managing (D) managed
>
>
> Example 3:
> > Question: Please complete the _______ below with your enquiry and a member of our team will contact you shortly.
> >
> > Options: (A) reformists (B) reforms (C) deformed (D) form

---

### Official Review · Reviewer_SEmd · 2023-08-07

**Soundness:** 4

**Excitement:**

4: Strong: This paper deepens the understanding of some phenomenon or lowers the barriers to an existing research direction.

**Paper Topic And Main Contributions:**

The paper concerns the task of generating explanations for answers to multiple choice cloze questions, especially those aimed at language learners, which tend to have grammar-based distractors and explanations. It introduces an automatically generated dataset based on extracting question and explanation patterns from existing tests, applying these patterns to unlabeled text using syntactic parsing, and diversifying the language by using large language models to paraphrase.

**Questions For The Authors:**

Line 490, "Despite the large amount of training data..." - why would acceptable fluency be in spite of training data? Would that help?

**Reasons To Accept:**

- The application is important and pertinent. The task of generating explanations for cloze tests would benefit language learners. It is apparently difficult enough that current models cannot master it, but conceivably could in the near future.
- A new dataset is introduced. Though it is not manually generated, it is based on manually generated data and uses reasonable data augmentation techniques.
- The methods for creating the dataset are interesting. They combine crafted rules incorporating linguistics and syntactic parsing but also use large language models in subtle and appropriate ways.
- Benchmark systems are rigorously evaluated, including by humans. Current relevant models are tried, including fine-tuned sequence-to-sequence and generative models and zero-shot instruction-following models. Large language models are also employed for evaluation, but with appropriate skepticism and validation, leading to further interesting results for their use in evaluation.


**Reasons To Reject:**

None.

**Reproducibility:**

5: Could easily reproduce the results.

**Reviewer Confidence:**

3: Pretty sure, but there's a chance I missed something. Although I have a good feel for this area in general, I did not carefully check the paper's details, e.g., the math, experimental design, or novelty.

---

> ### Author Rebuttal · Authors · 2023-08-28
>
> Thank you for the detailed feedback.
>
> In Line 490, we meant despite the LLMs could yield fluent text, generated text is not suitable for explaining the question. In other words, the LLMs we used are all pre-trained with a large amount of data and have the capability to generate fluent text for any given instruction without fine-tuning. However, in spite of the generated text being fluent, it is not valid as an explanation.
>
> We apologize for the ambiguous expression, we will modify it with a clear expression like: “Although LLMs are capable of generating text with acceptable fluency thanks to the large amount of pre-training data, they received low evaluations in terms of producing valid explanations.”

---

### Meta-Review · Area_Chair_YPuQ · 2023-09-10

**Recommendation:** 4

**Metareview:**

This paper presents a new task (and corpus) ClozEx which focuses on generating explanations for cloze question in language assessment with a focus on focus on English as a Second Language, ESL, learners. Various models have been fine-tuned to generate explanations for the questions. Extensive evaluations have been conducted in order to answer to two questions: (i) Do LLMs Explain Cloze Questions Well?, (ii) Are Automatic Metrics Reliable in ClozEx?.  The experiments are solid and the analysis interesting.
Some extra information could be provided concerning the annotators and the bias management.

---

### Decision · Program_Chairs · 2023-10-07

**Decision:**

Accept-Findings

**Comment:**

This paper presents a new task (and corpus) ClozEx which focuses on generating explanations for cloze question in language assessment with a focus on focus on English as a Second Language, ESL, learners. Various models have been fine-tuned to generate explanations for the questions. Extensive evaluations have been conducted in order to answer to two questions: (i) Do LLMs Explain Cloze Questions Well?, (ii) Are Automatic Metrics Reliable in ClozEx?.  The experiments are solid and the analysis interesting.
Some extra information could be provided concerning the annotators and the bias management.